# New PMMA-Based Hydroxyapatite/ZnFe_2_O_4_/ZnO Composite with Antibacterial Performance and Low Toxicity

**DOI:** 10.3390/biomimetics8060488

**Published:** 2023-10-14

**Authors:** Olga Bakina, Natalia Svarovskaya, Ludmila Ivanova, Elena Glazkova, Nikolay Rodkevich, Vladyslav Evstigneev, Maxim Evstigneev, Andrey Mosunov, Marat Lerner

**Affiliations:** 1Institute of Strength Physics and Material Science, Siberian Branch of Russian Academy of Science, Av. Akademicheskii, 2/4, 634055 Tomsk, Russia; nvsv@ispms.ru (N.S.); eagl@ispms.tsc.ru (E.G.); ngradk@ispms.tsc.ru (N.R.); lerner@ispms.tsc.ru (M.L.); 2Sevastopol State University, 33 Universitetskaya Street, 299053 Sevastopol, Russia; vald_e@rambler.ru (V.E.); max_evstigneev@mail.ru (M.E.);

**Keywords:** polymethylmethacrylate, antimicrobial nanoparticles, electrical explosion of wire, hydroxyapatite

## Abstract

Polymethylmethacrylate (PMMA) is the most commonly used bone void filler in orthopedic surgery. However, the biocompatibility and radiopacity of PMMA are insufficient for such applications. In addition to insufficient biocompatibility, the microbial infection of medical implants is one of the frequent causes of failure in bone reconstruction. In the present work, the preparation of a novel PMMA-based hydroxyapatite/ZnFe_2_O_4_/ZnO composite with heterophase ZnFe_2_O_4_/ZnO NPs as an antimicrobial agent was described. ZnFe_2_O_4_/ZnO nanoparticles were produced using the electrical explosion of zinc and iron twisted wires in an oxygen-containing atmosphere. This simple, highly productive, and inexpensive nanoparticle fabrication approach could be readily adapted to different applications. From the findings, the presented composite material showed significant antibacterial activity (more than 99% reduction) against *P. aeruginosa*, *S. aureus*, and MRSA, and 100% antifungal activity against *C. albicans,* as a result of the combined use of both ZnO and ZnFe_2_O_4_. The composite showed excellent biocompatibility against the sensitive fibroblast cell line 3T3. The more-than-70% cell viability was observed after 1–3 days incubation of the sample. The developed composite material could be a potential material for the fabrication of 3D-printed implants.

## 1. Introduction

Currently, polymethylmethacrylate (PMMA) is the most commonly used bone void filler in orthopedic surgery [1]. It is particularly common in percutaneous vertebroplasty in cases of vertebral compression fracture [2]. PMMA has a high stiffness and chemical stability compared to other acrylic polymers [3]. However, the biomedical applications of PMMA are limited by its bioinertness: the polymer is not suitable for bone ingrowth [4]. The adhesion between PMMA and the adjacent bone tissue is usually low, which is the main reason for the rejection of PMMA implants. In addition, the exothermic reaction that occurs during its curing can also cause bone necrosis [5]. The poor antibacterial properties of PMMA significantly limit its application as a prosthetic material [6]. To improve biocompatibility, PMMA-based composites containing calcium-phosphate ceramics with the necessary biocompatibility are most often developed [7].

Hydroxyapatite (HA) is the most common ceramic for biomedical applications due to its chemical composition being close to human bones and teeth [8]. HA is known to have osteoconductive and osteophilic properties. Previous studies have shown the ability of HA to stimulate osteointegration [9] and osteoblast proliferation [10]. Consequently, PMMA–HA composites have a good ability to create a strong bond between the bone and prosthesis [11,12].

In addition to insufficient biocompatibility, the infection of medical implants is one of the frequent causes of failure in bone reconstruction [13]. PMMA/HA-based composites with antibacterial properties may be a solution to this problem. Antibiotics are a common solution, but they have serious side effects that cannot be ignored [14], including biofilm formation and the emergence of bacterial resistance. HA composites with silver nanoparticles can effectively inhibit the growth of bacteria and fungi [15,16,17]. However, when applying silver to nanocomposites, cell-growth problems arise due to its cytotoxicity [18]. In addition, osteoblast adhesion has been shown to be decreased when increasing the silver content, as does cell viability [19].

In this regard, new antimicrobial agents with a low toxicity, based on nanoparticles (NPs) of biologically active metals such as iron, zinc, copper, etc., are actively pursued. The antimicrobial activity of ZnO NPs is provided by their ability to generate reactive oxygen species (ROS)-damaging components in the bacterial cell. A summary of the data on the minimum inhibitory concentration of NPs against the main pathogens of purulent infection recently reviewed by E. Sanchos-Lopez et al. [20], showed that ZnO NPs inhibit bacterial growth in concentrations starting from only 40 µg/mL. The modification of ZnO NPs with other elements is a promising approach to improve their antibacterial properties. A promising but insufficiently investigated class of compounds with antimicrobial properties is spinel ferrites. Currently, prospects for their use as contrast agents for magnetic resonance imaging [21], targeting agents and substances for magnetic hyperthermia [22] have been shown. Among spinel ferrites, ZnFe_2_O_4_ NPs attract special attention as antimicrobial agents due to their bio-friendly character, lower toxicity than other metal ferrites, chemical stability, easy and reproducible synthesis, low saturation, and unique magnetic properties [23]. However, their antibacterial activity is low. 

The use of bicomponent ZnO/ZnFe_2_O_4_ NPs to create antibacterial agents can lead to the more effective suppression of bacterial growth due to the synergistic effect of nanoparticle components. This approach allows one to reduce the content of the active ingredient and, consequently, its toxicity, increasing its antibacterial efficacy. Thus, coral-like ZnO/C-ZnFe_2_O_4_ (Z/C-ZFO) NPs were found to possess sterilization efficiencies towards *E. coli* and *S. aureus* bacteria of 99.4% and 98.0%, respectively, combined with a super cycling stability [24]. Magnetic hybrid ZnO/ZnFe_2_O_4_ nanocomposites with an average crystallite size of 30 nm, obtained using a co-precipitation method, also effectively inhibited the growth of *E. coli* and *S. aureus* pathogens with efficiencies of 98.6 and 97.4%, respectively [25]. ZnO/ZnFe_2_O_4_ nanoparticles obtained through the electrical explosion of two wires completely inhibited the growth of MRSA [26]. Unlike the contrasting agents commonly used, magnetic nanoparticles ranging in sizes from 30 to 150 nm are able to circulate in the blood for a long time, which reduces their side effects associated with nephrotropicity and hepatotrophicity [27]. Thus, PMMA-based composites containing HA and ZnO/ZnFe_2_O_4_ NPs can provide increased cell adhesion, and impart antibacterial activity and radiopacity to the composites. In the present work, the preparation of a novel PMMA-based hydroxyapatite/ZnFe_2_O_4_/ZnO composite with heterophase ZnFe_2_O_4_/ZnO NPs as an antimicrobial agent was described. This composite material could be a potential material for the fabrication of 3D-printed bone scaffolds and connecting screws. However, no work on obtaining such composites has yet been published.

## 2. Materials and Methods

### 2.1. Synthesis of ZnFe_2_O_4_/ZnO Heterophase Nanoparticles

The simultaneous electrical explosion of iron and zinc twisted wires (EETW) in an oxygen-containing atmosphere was used to produce antibacterial heterophase NPs ZnFe_2_O_4_/ZnO. This method is of industrial importance due to the use of metal wires as precursors of high productivity (at least 200 g per hour). The electrical explosion was carried out in a gas mixture of argon + 20 vol% oxygen. This argon/oxygen ratio ensured the complete oxidation of the metals, as has previously been established [28]. The Fe and Zn wires were 0.17 mm and 0.39 mm in diameter, respectively, and 75 mm in length. This metal ratio in the twist gives NPs their highest antibacterial activity [26]. The electrical explosion was carried out using a UDP-2 (Advanced Powder Technologies LTD, Tomsk, Russia) device at a voltage of 26 kV, and an energy storage capacity of 2.4 µF in the fast explosion mode, which was realized when a current pulse with a density of 10^7^−10^8^ A/cm^2^ flowed through the wires. A schematic diagram of nanoparticle and composite production is shown in Figure 1. A detailed description of the production of NPs is given in [26].

### 2.2. Fabrication of PMMA-Based HA/ZnFe_2_O_4_/ZnO Composite

To obtain a PMMA-based HA/ZnFe_2_O_4_/ZnO composite, polymethyl methacrylate pellets (Diapolyacrylate Co., Ltd., Rayong, Thailand), with a mean M_w_~120,000, and HA powder (Natural Works Ingeo 40–43 d, Nature Works LLC, Minneapolis, MN, USA) were thoroughly mixed with ZnFe_2_O_4_/ZnO nanoparticles in concentrations of 5 wt% and 7 wt%. Based on preliminary studies, it was found that the addition of nanoparticles in an amount of less than 5% did not provide enough antimicrobial activity in the composite. A solution of PMMA in acetone (10% wt.) was pre-prepared for uniform mixing. The ZnFe_2_O_4_/Zn NP and HA powders were gradually introduced into the solution. The resulting mixture was thoroughly dispersed using an ultrasonic bath with a frequency of 40 ± 2 kHz (WU-09-«Y-FP»-03, Yaroslavl, Russia). After removing the solvent, the resulting mixture was pulverized and pellets were obtained. The mixture in the form of cut pellets was then loaded into a Rondol twin-screw extruder (Twin-Screw Extruders, Microlab, Shenzhen, China) to obtain pellets with a diameter of 1.75 mm. Further, the samples required for the studies were printed from the obtained pellets using a Duplicator i3 laboratory printer (Wanhao, China). The extruder nozzle diameter was 0.4 mm, and the temperature at the extrusion head outlet was 210 °C. As a result, a number of samples were prepared, listed in Table 1.

### 2.3. Characterization

The nanoparticles and composite samples were characterized using XRD (XRD 6000, diffractometer, Shimadzu Corporation, Kyoto, Japan), SEM (LEO EVO 50, Germany) and transmission electron microscopy (TEM) (JEM JEOL 2100, Japan), and nitrogen adsorption/desorption (Sorbtometer M, Katakon, Russia). The particle size was determined using TEM image analysis (≈1700 particles); the size of the particle agglomerates was determined using the sedimentation method using a CPS 24000 disc centrifuge (CPS Disk Instruments, Prairieville, LA, USA). The magnetic properties of the nanoparticles were studied using a JR-6 spinner magnetometer (Agico, Brno, Czech Republic). Zeta potential measurements were performed in deionized water at 25 °C and various pH values using a Zetasizer Nano ZSP instrument (Malvern, UK). The surface composition was investigated using X-ray photoelectron spectroscopy (SPECS Surface Nano Analysis GmbH, Berlin, Germany). The release of iron and zinc cations from the composite was studied when exposing the composite samples in NaCl solution (0.9%) at 37 °C for 30 days. The cation concentration was determined using the inversion voltammetry method (TA-2, TomAnalit, Tomsk, Russia).

### 2.4. Antibacterial and Antifungal Activity of PMMA-Based HA/ZnFe_2_O_4_/ZnO Composite

The antibacterial activities of the samples were assessed using the droplet contamination method according to ISO 22196:2011, with the following three strains of bacteria: *Staphylococcus aureus* ATCC 6538-P (*S. aureus*), *Pseudomonas aeruginosa* ATCC 27853 (*P. aeruginosa*) and methicillin-resistant *Staphylococcus aureus* ATCC 43300 (MRSA). *Candida albicans* (ATCC SC5314 strain) was cultured at 37 °C in Sabouraud agar (Merck, Rahway, NJ, USA). 

Three samples of each sample type and three control samples were used. The samples were placed in a sterile 24-well plate. A volume of 64 μL (1 µL per 1 mm^2^ sample area) of the bacterial suspension with a concentration 10^5^ CFU/mL was placed applied to the samples surface and spread evenly on the surface using a spatula sterile swab. As a control, three empty well plates were inoculated with 64 μL of the test culture suspension on the surface of titanium samples without a coating (to form control samples). The exposure time was 6 h. A further increase in the exposure time led to a decrease in the viability of bacteria in the control sample. After 6 h of exposure, samples were placed in sterile containers containing 10 mL of neutralizing agent (0.9% wt.% sodium chloride solution, Grotex, St-Petersburg, Russia) and shaken for 10 min using an orbital shaker PSU-20i (SIA Biosan, Latvia) to remove adhered bacteria. From each supernatant, 100 µL was aliquoted onto Mueller–Hinton agar (HiMedia Laboratories, Thane, India) plates. Then, the liquid supernatant was diluted by factors of 10, 100 and 1000 and also spread onto Mueller–Hinton agar plates (Sabouraud agar plates for *C. albicans*). The prepared plates were subsequently incubated at 37 ± 1 °C for 24 h. The colony plate count method was used for the enumeration of CFUs after incubation. The antibacterial rate (*AR*) of each sample was evaluated by calculating the percentage (%) reduction in microbial contamination of the test samples compared to the control samples, according to the following equation:(1)$$AR=\frac{N_{0}-N_{x}}{N_{0}}\times 100$$
where *N*_0_ is the number of colonies corresponding to the blank control sample, and *N_x_* is the number of colonies corresponding to the composites.

### 2.5. Biocompatibility of PMMA-Based HA/ZnFe_2_O_4_/ZnO Composite

The MTT and flow cytometry tests were used to investigate the biocompatibility of the composite samples. The mouse embryonic fibroblast cell line NIH/3T3 was used to evaluate the cell compatibility of the samples. For this, cells were cultured within 24 h in Dulbecco’s Modified Eagle Medium (HiClone, San Angelo, TX, USA) with the addition of 10% fetal bovine serum (HiClone, USA) and 1% penicillin/streptomycin (Biolot, Russia) at 37 °C and 5% CO_2_ (Sanyo, Tokyo, Japan).

Biocompatibility of the samples was investigated using the direct contact method. For this purpose, composite samples were placed in the wells of a 24-well plate, on the surface of which a cell suspension was added in the amount of 50,000 cells per sample. To improve the adhesion of cell culture to the substrate, composite samples with inoculum were placed in the incubator for 3−5 min. Further, 1 mL of nutrient medium was added to each well. The cells on the surface of the samples were incubated for 24 h at 37 °C in an atmosphere of 5% CO_2_. After incubation, cell viability was examined.

For cell-viability examination using the MTT test, cells were detached from the surface of the composite samples and 100 μL cell suspensions were each transferred into 96-well plates. A volume 10 µL of MTT solution (3-(4.5-dimethyl thiazol-2-yl)-2.5-diphenyl-tetrazolium bromide) was added to each well with cells. Incubation with MTT solution was at 37 ± 1 °C for 2 h in an atmosphere of 5% CO_2_. At the end of incubation, the nutrient medium was carefully removed and 100 μL of dimethyl sulfoxide (Biolot, Russia) was added to each well to dissolve formazan crystals. After 15 min, the optical density was determined on a Multiscan FC microplate spectrophotometer (Termo Scientific, Bremen, Germany) at a wavelength of 570 nm. Then, the percentage of living cells (viability, %) was calculated. For the method of flow cytometry, cells after incubation were washed with Dulbecco’s phosphate-buffered saline DPBS (Biolot, Russia) and harvested using 0.25% trypsin-EDTA (HiClone, USA). Cells were counted using a TC20 Automated Cell Counter (Bio-Rad, Hercules, CA, USA). Cells (70,000 cells per well) were cultured in 24-well plates at 37 °C for 24 h in the presence of samples. After incubation, cells were washed with DPBS and harvested with trypsin-EDTA. Normal and apoptotic cells were distinguished using a FITC Annexin V Apoptosis Detection Kit with 7-AAD (BioLegend, France) in a Cytoflex flow cytometer (Beckman Coulter, Brea, CA, USA).

## 3. Results and Discussion

### 3.1. Preparation and Physicochemical Characteristics of ZnFe_2_O_4_/ZnO NPs

The NPs were of a predominantly spherical and faceted shape (Figure 2a). Particles larger than 500 nm were encountered in the sample, but their number was small. Energy-dispersive X-ray Spectroscopy (EDS) analysis showed that zinc, iron, and oxygen were uniformly distributed throughout the volume of the particles, and their signals were proportional to the fraction of metals (Figure 2b), which indicated the heterophase structure of the particles and fairly uniform mixing of metals during EETW synthesis, and was consistent with the ideas about the mechanism of nanoparticle formation during EETW process [29].

The obtained NPs followed a log-normal size distribution; the average size of the NPs was 87 nm (Figure 3a). According to sedimentation analysis in the density gradient, the particles were mostly agglomerated, the average size of the NPs agglomerates being 0.2–0.5 μm (Figure 3b). The NP-specific surface area was 13.7 m^2^/g. From the rough approximation of the particle density ρ about 5 g/cm^3^, calculation of the average particle size gave a value of about 90 nm, which agreed well with the electron microscopy data and indicated the non-porous structure of the particles.

XRD phase analysis (Figure 4) showed that the phase composition of the nanoparticles obtained was close to that characteristic of the nanoparticles, with the same component ratio synthesized in the equilibrium conditions. Zinc oxide ZnO (JCPDS# 96-900-4179) with hexagonal wurtzite structure (≈81.3%) and zinc ferrite ZnFe_2_O_4_ (JCPDS# 96-900-2489) with cubic spinel structure (≈10.7%) phases were detected in the NPs. Zinc in a zero oxidation state (JCPDS# 96-900-8523) was also present in the sample in an amount of approximately 8%.

For detailed analysis of the of heterophase ZnFe_2_O_4_/ZnO NP surface, the spectra were decomposed into individual components. The obtained Fe*2p* spectrum was a Fe*2p*_3/2_-Fe*2p*_1/2_ doublet with a 2:1 ratio of component integral intensities (Figure 5a). The position of the Fe*2p*_3/2_ main line, the intensity and relative position of the “shake-up” satellite lines in the Fe*2p* spectrum were used to determine the iron state. For ZnFe_2_O_4_/ZnO NPs, the Fe*2p*_3/2_ spectrum was a peak with binding energy around 710.8 eV. In addition, a “shake-up” satellite was observed, which was 8.7 eV away from the main peak. The large value of the binding energy and the presence of “shake-up” satellites allowed to state that iron is in the Fe^3+^ state in this sample. The Zn*2p* spectrum was represented by a Zn*2p*_3/2_−Zn*2p*_1/2_ duplet with a spin-orbit splitting of 23.1 eV, which arises from the spin–orbit interaction and splitting of the *2p*-level of Zn (Figure 5b). The shape and position of the Zn*2p*_3/2_ line, as well as the values of the binding energy of the Zn*2p*_3/2_ level in the range of 1021.5−1022.0 eV was consistent with Zn^2+^. Thus, both zinc(II) and Fe(III) were found to be present on the surface of NPs.

The surface potential of biomaterials is a key factor regulating cell responses, driving their adhesion and signaling in tissue regeneration [30]. The most common parameter to quantitatively describe the surface potential of biomaterials that are in contact with a liquid is zeta (ζ) potential. The zeta potential is the potential of the electrical double layer that occurs at the solid–liquid interface. The measurement of the zeta potential is often crucial for the design of biomaterial surfaces [30]. In addition, the zeta potential is one of the main quantitative characteristics of the charge properties of NPs, determining the structure of the electric double layer and regulating the stability of aqueous suspensions of nanoparticles, as well as the nature of their physical interaction with other objects (cells, bacteria, polymers, etc.). The zeta potential of ZnFe_2_O_4_/ZnO NPs was calculated from the NP electrophoretic mobility and determination of the boundary velocity using dynamic light scattering. Figure 6 shows the zeta-potential variation curve of ZnFe_2_O_4_//ZnO NPs as a function of pH. ZnFe_2_O_4_/ZnO NP nanoparticles at physiological pH = 7 had a positive zeta potential of 29.8 ± 0.4 mV, which was close to the zeta potential of the ZnO nanoparticles [31]. The presence of a positive zeta potential would favor the interaction of the NPs with the negatively charged bacterial-cell surface. The isoelectric point (IEP) of the NPs was 9.22. IEP is known to be directly related to cell responses [32]. For example, the IEP for collagen is 9.3, which is close to that of ZnFe_2_O_4_/ZnO. Thus, the presence of NPs is expected to have a positive effect on cell adhesion.

Data on the field dependence of the magnetization of nanoparticles is given in Figure 7. 

The magnetization value at 300 K for an applied field of 1 Tesla was approximately 3.2 emu/g. This value is comparable to that obtained by Xue et al., who also measured a magnetic ZnO/ZnFe_2_O_4_/diatomite composite at 300 K, synthesized using a hydrothermal–precipitation method [33]. In addition, a magnetic hysteresis loop was clearly observed in the weak field region, as shown in the inset of Figure 7.

The saturation magnetization (MS) of the composite nanoparticles (3.2 emu/g) was less than that of pure zinc ferrite (28.4 emu/g [34] и 8 emu/g [35]). The main reason for the decrease in the MS value in the present work is that mixing zinc ferrite with a non-magnetic phase (ZnO) reduces the total magnetization of the composite [36]. In addition, the reduction in saturation magnetization can also come from multiple magnetic domains in one composite, and aggregates contain multiple particles. Interactions between the domains and close particles can cause a reduction in the saturation magnetization. However, the synthesized NPs have enough of a magnetic property to ensure radiopacity of the HA/PMMA-based composites.

### 3.2. Characterization of PMMA-Based HA/ZnFe_2_O_4_/ZnO Composite

Figure 8 shows SEM images of the fracture surfaces of the HA/PMMA-based composites, with different contents of ZnFe_2_O_4_/ZnO NPs at different magnifications. As can be seen in the figures, the morphology of the composite samples did not depend on the NP content; the HA particles are uniformly dispersed in the PMMA matrix. These magnifications do not allow the visualization of the nanoparticles; however, it can be seen from the SEM-EDS elemental analysis data that iron and zinc were uniformly distributed throughout the composite sample, suggesting a uniform distribution of nanoparticles. An increase in the mass fraction of nanoparticles leads to an increase in the number of agglomerates (marked with arrows in Figure 8b).

The release of iron and zinc ions was investigated under the conditions of the exposure of composite samples in an NaCl physiological solution (0.9%) at 37 °C for 30 days, with these conditions simulating the cellular fluid (Figure 9). The ratio of the surface area to the solution volume was 1 cm^2^/50 mL, according to the ASTM G31-21 recommendations.

The use of the NaCl solution as a model cellular fluid can promote the release of metal cations during the dissolution of NPs located on the composite surface. For the PMMA/HA/NP (5%) and PMMA/HA/NP (7%) composite samples, the maximum content of cations released was found to be 0.029 ± 0.002 mg/L and 0.034 ± 0.005 mg/L for the Zn^2+^ and Fe^3+^ cations, respectively. That is, all the cations were released in micro quantities, and the concentrations of Fe^3+^ cations (0.3 mg/L) and Zn^2+^ cations (5 mg/L) were much lower than even the maximum permissible concentrations in drinking water. According to the values of the inhibitory concentrations of iron and zinc ions against, for example, *S. aureus*, the greatest contribution to the composite’s antimicrobial activity is made by the zinc ions. The authors of [37] showed that Zn^2+^ ions effectively kill *S. aureus* at a concentration of 10^−4^ M (0.007 mg/L). At the same time, Fe^3+^ ions inhibited the growth of *S. aureus* ATCC 25923 at a concentration of 1 mM (56 mg/L) by 80% [38]. 

Due to low specific gravity and electron density, PMMA-based implants cannot be detected using computed tomography (CT) and magnetic resonance imaging, making it impossible to assess the position and condition of the device after placement. As can be seen from the typical images in Figure 10, the specimen of pure PMMA is invisible under computed tomography (Figure 10a), whereas the presence of nanoparticles significantly increased the contrast of the images (Figure 10b). Even the samples with an NP content of 5 wt% could be easily identified.

### 3.3. Antimicrobial Activity

Figure 11 shows the effect of the synthesized samples on the viability of *S. aureus, P. aeruginosa* and MRSA bacterial strains, as well as *C. albicans* fungi. The antibacterial activity of the PMMA/NP composites was previously investigated by us in the article [39]. It was found that a mass fraction of nanoparticles in the composite of less than 5% weakly inhibited bacterial growth. This effect was due to the small amount of antibacterial nanoparticles on the surface of the composite.

As can be seen from the presented data, the composite samples (PMMA/HA/NPs) showed significant antimicrobial and antifungal effects in contrast to the samples without nanoparticles (PMMA and PMMA/HA). The PMMA/HA/NP (7%) composite showed almost complete growth suppression of all the microorganisms studied (AR more than 99.9%). The data obtained confirm the crucial contribution of NPs to the antimicrobial activity of the composites studied. It was previously shown that the bacterial activity of the ZnFe_2_O_4_/ZnO NPs is due to two mechanisms:(1)The generation of reactive oxygen species O_2_^–^• и OH• [24,40], which can cause the destruction of cell membranes, proteins, and DNA through oxidation, and eventually cause bacterial inactivation;(2)Positive charges from NPs are attracted to the cell surface by electrostatic interactions, and the difference in the electrostatic gradient leads to damage on the cell surface of ZnO NPs [41,42];(3)Teicoic and lipoteichoic acids act as a chelating agent on Zn^2+^ ions, which are then carried by passive diffusion across membrane proteins [41].

Figure 12 shows SEM images of the surface of the PMMA and PMMA/HA/NP (7%) samples with *S. aureus* bacteria after 6 h of incubation. No bacteria were detected on the surface of the samples with NPs after 24 h of incubation. After 6 h incubation with the composite sample, the bacteria surface became wrinkled and rough (Figure 12a), and some of the bacteria were completely destroyed (Figure 12a, indicated by arrows). At the same time, the control *S. aureus* incubated with the PMMA sample had a smooth surface without defects (Figure 12b). Thus, the SEM images demonstrate bacterial morphological changes after the composite-sample treatment, including a broken cell membrane, loss of normal appearance, and wrinkled and rough surface. Similar phenomena have also been observed in other *S. aureus* experiments, caused by silver nanoparticles [43,44] and graphene oxide/cobalt ferrite nanoparticles [45]. The results clearly indicated the effect of NPs on the integrity of the bacterial cells.

Thus, nanoparticle modification significantly improves the antibacterial properties of the composite material, and the uniform distribution of NPs in the PMMA matrix contributes to this sustained antibacterial effect upon wear during clinical use.

### 3.4. Biocompatibility of PMMA-Based HA/ZnFe_2_O_4_/ZnO Composite

Although PMMA and HA are considered non-toxic materials, the biotoxicity of their additives in the form of nanoparticles is a matter of serious concern. For example, BaSO_4_ particles induced osteolysis when administered as a contrast agent [46]. An MTT test and live–dead assay using flow cytometry were performed to evaluate the cytocompatibility of the composite samples (Figure 13). Cells of the sensitive fibroblast cell line 3T3 were cultured on the composite samples. No toxic effect on the cells was observed (>70% cell viability) after 1–3 days. A weak toxic effect was demonstrated only by the PMMA sample (cell viability ≈ 90%), which was in agreement with the PMMA toxicity data reported [47,48] for a number of cell lines.

The HA-containing samples provided similar or higher cell viability compared to the control (cells in growth medium only). The data obtained show that composite samples containing NPs are not only non-toxic to cells, but also enhance fibroblast proliferation. Calcium phosphate additives can effectively bind serum proteins and growth factors, stimulating cell attachment and proliferation [46]. In addition, calcium ions form positive charge sites [49] that support the adsorption of key proteins that affect cell adhesion [50]. 

## 4. Conclusions

In summary, this study has presented a multi-functional composite material with a strong potential for use in implant applications. The PMMA matrix was decorated with two types of additive (HA and ZnO/ZnFe_2_O_4_ nanoparticles) to provide antibacterial and antifungal characteristics, biocompatibility and radiopacity. The simple, highly productive, and inexpensive nanoparticle fabrication approach could be readily adapted to different applications. From the findings, the presented composite material showed significant antibacterial activity (more than 99% reduction) against *P. aeruginosa*, *S. aureus*, and MRSA, and 100% antifungal activity against *C. albicans,* as a result of the combined use of both ZnO and ZnFe_2_O_4_. In vitro cytotoxicity assay using a 3T3 fibroblast cell line showed that all composites were biocompatible, showing >90% cell viability. In addition, for further clinical application, the in vivo properties of the composite material need to be further studied.

## Figures and Tables

**Figure 1 biomimetics-08-00488-f001:**
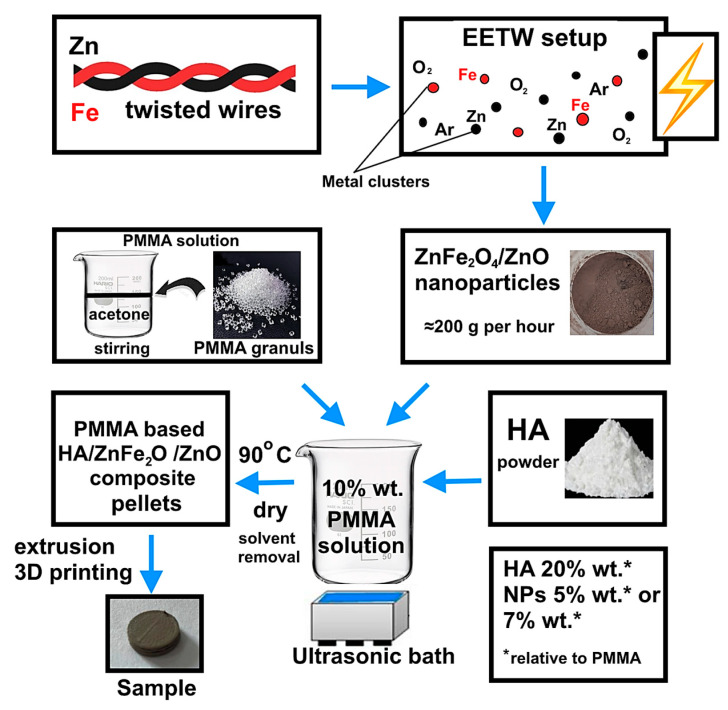
Scheme of NPs and composite preparation.

**Figure 2 biomimetics-08-00488-f002:**
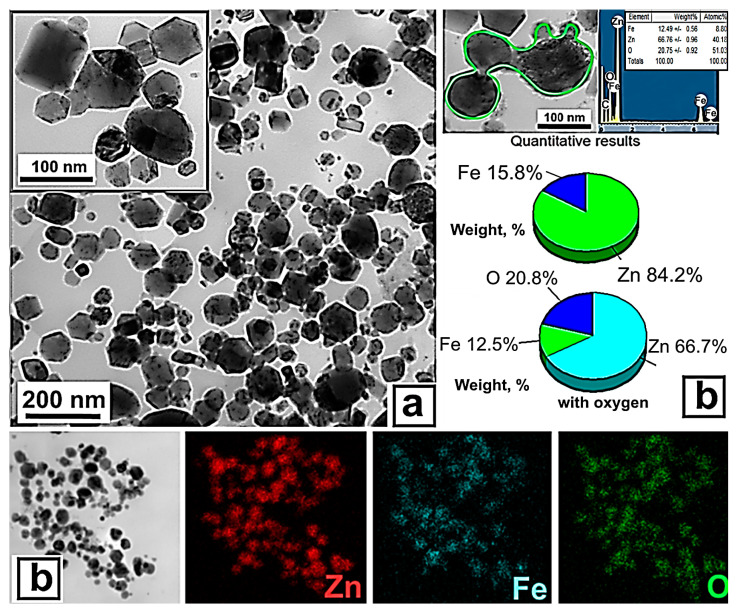
TEM image (**a**) and EDX analysis (**b**) of ZnFe_2_O_4_/ZnO NPs.

**Figure 3 biomimetics-08-00488-f003:**
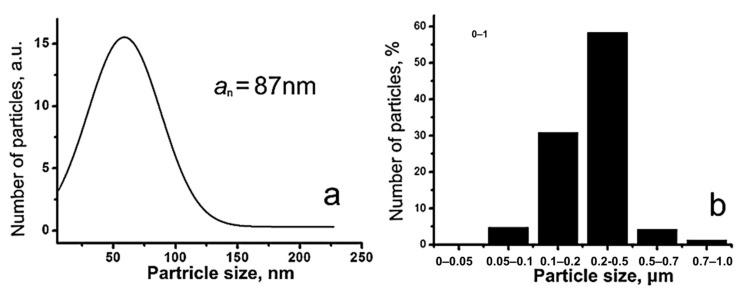
ZnFe_2_O_4_/ZnO nanoparticle (**a**) and agglomerate (**b**) size distribution.

**Figure 4 biomimetics-08-00488-f004:**
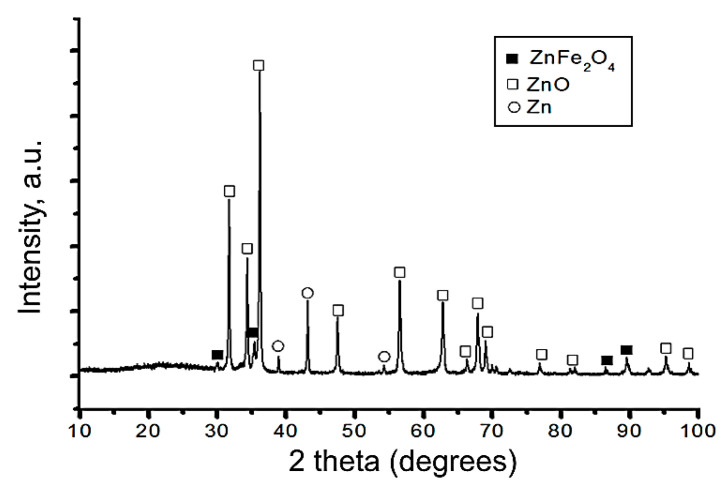
XRD diffractogram of the ZnFe_2_O_4_/ZnO nanoparticles.

**Figure 5 biomimetics-08-00488-f005:**
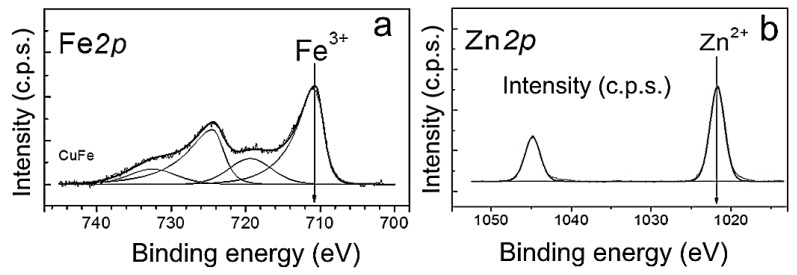
X-ray photoelectron spectra of Fe*2p* (**a**) and Zn*2p* (**b**) for ZnFe_2_O_4_/ZnO NPs. The spectra were normalized by the integral intensity of the peaks corresponding to the metals included in the NPs.

**Figure 6 biomimetics-08-00488-f006:**
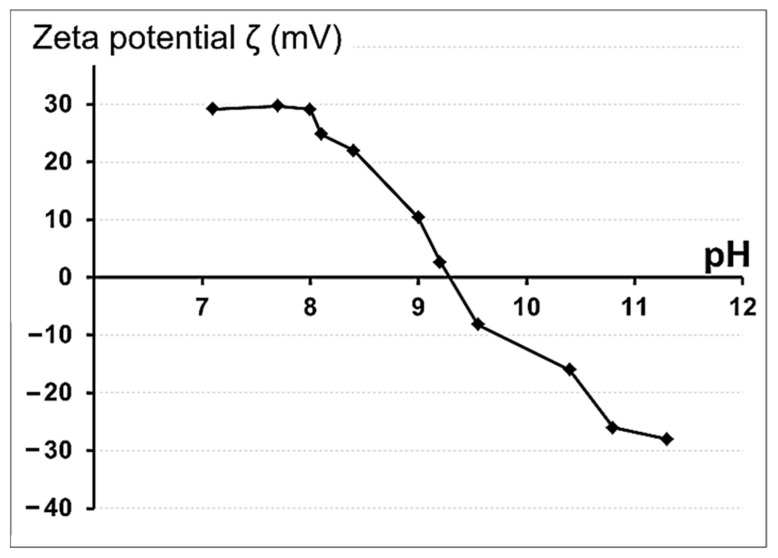
Surface zeta potential of ZnFe_2_O_4_/ZnO NPs. NPs displayed as a function of pH.

**Figure 7 biomimetics-08-00488-f007:**
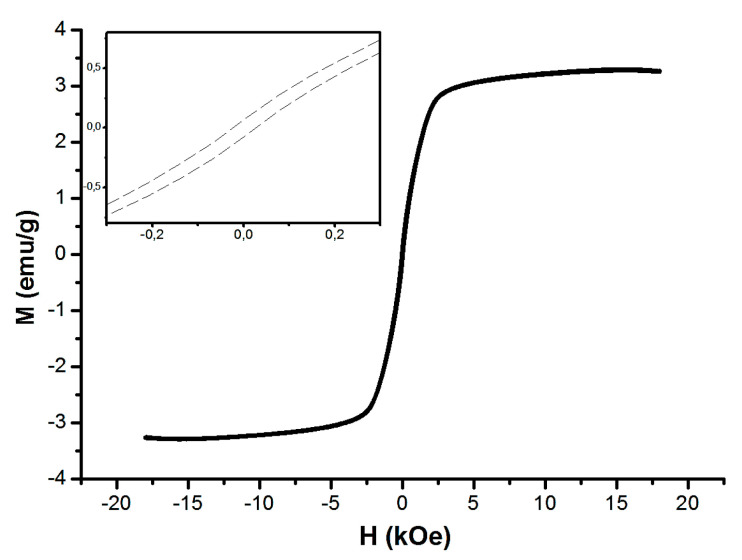
Field dependence of the magnetization and magnetic hysteresis loop (inset) at 300 K for ZnFe_2_O_4_/ZnO NPs.

**Figure 8 biomimetics-08-00488-f008:**
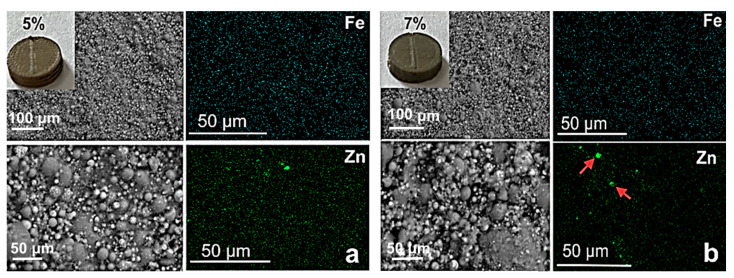
SEM images and EDX analysis of PMMA/HA/NP (5%) composite (**a**) and of PMMA/HA/NP (7%) composite (**b**). NP agglomerates are marked by red arrows.

**Figure 9 biomimetics-08-00488-f009:**
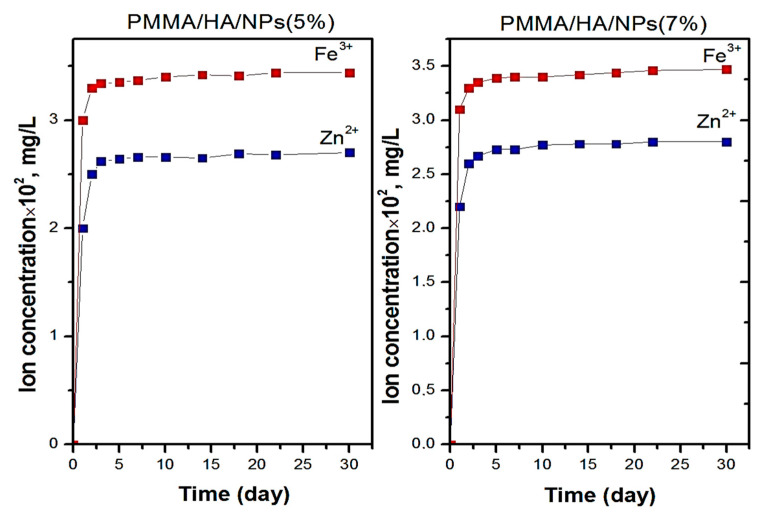
Zinc- and iron-ion release for PMMA/HA/NP composites.

**Figure 10 biomimetics-08-00488-f010:**
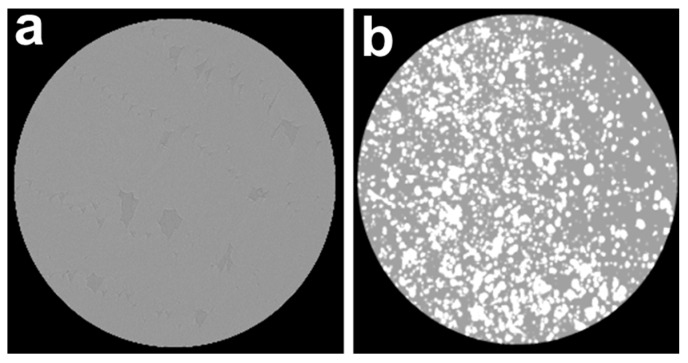
CT images of (**a**) PMMA and (**b**) PMMA/HA/NP (5%) specimens.

**Figure 11 biomimetics-08-00488-f011:**
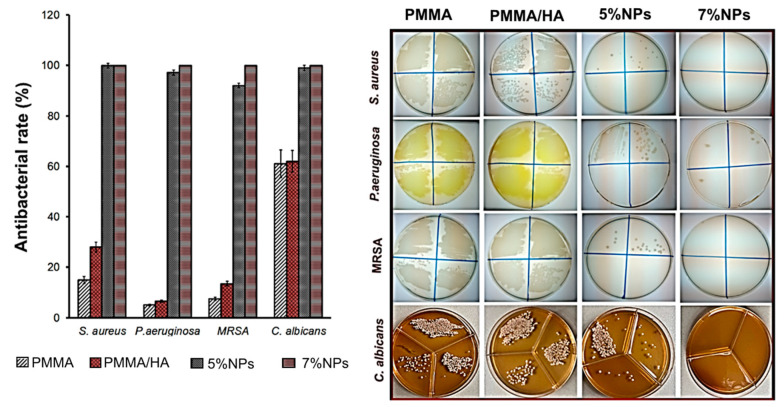
Antibacterial and antifungal activity of composite samples. Samples 5% NPs and 7% NPs are PMMA/HA/NPs (5%) and PMMA/HA/NPs (7%), respectively.

**Figure 12 biomimetics-08-00488-f012:**
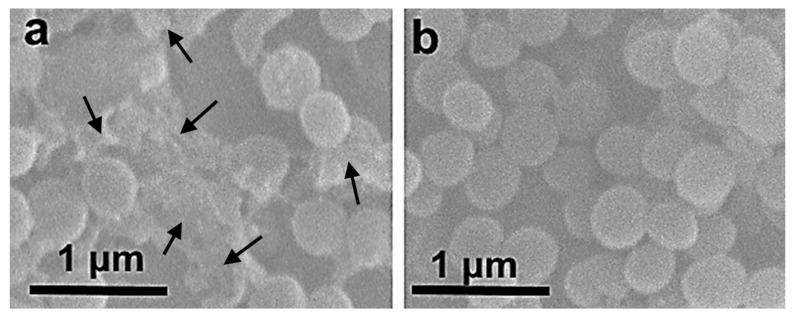
SEM images of *S. aureus* after 6 h incubation on the (**a**) PMMA/HA/NP- (7%) and (**b**) PMMA-sample surface.

**Figure 13 biomimetics-08-00488-f013:**
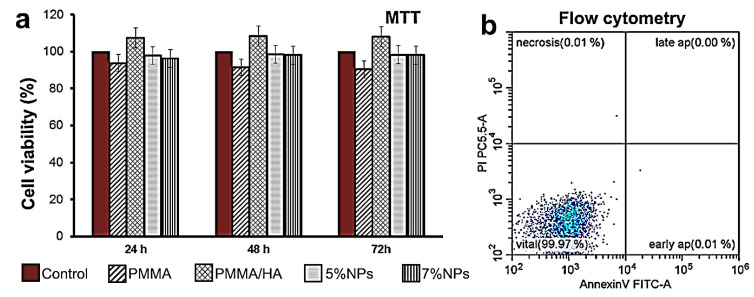
Cell viability of 3T3 cells co-cultured with samples, estimated using MTT test (**a**), and apoptosis and necrosis of 3T3 cell line analyzed using Annexin V/propidium iodide double staining followed by incubation for 24 h with PMMA/HA/NP (7%) composite sample (**b**). Samples 5% NPs and 7% NPs are PMMA/HA/NPs (5%) and PMMA/HA/NPs (7%), respectively.

**Table 1 biomimetics-08-00488-t001:** Detailed description of synthesized samples prepared.

Sample Designation	Component Mass Ratio, %		
	PMMA	HA	ZnFe_2_O_4_/ZnO NPs
PMMA	100	0	0
PMMA/HA	80	20	0
PMMA/HA/NPs (5%)	75	20	5
PMMA/HA/NPs (7%)	73	20	7

## Data Availability

Not applicable.

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
