# Peer review of "New PMMA-Based Hydroxyapatite/ZnFe2O4/ZnO Composite with Antibacterial Performance and Low Toxicity"

_biomimetics, 2023, doi:10.3390/biomimetics8060488_

Round 1
Reviewer 1 Report
It can be seen that the authors layout the extensive work with PMMA, HA, and embedded with ZnFe2O4/ZnO. The comments for this manuscript are as follows :
1. It is suggested the sub-title for 2.2 is changed to Fabrication instead of Synthesis..
2. How is the homogeneity of the mixture between PMMA, HA and ZnFe2O4/Zn is obtained?
3. What is the basis of selecting the mass percentage of ZnFe2O4/Zn (5% and 7%)?
4.Based on the Figure 8, ion released for Fe3+ is higher than Zn2+, therefore which elemental is mostly induced the antimicrobial activities?
5.Figure 9- label (a) and (b) is not included in the figure
6. Figure 10- the legend is not clear
7. Figure 11- please locate any significant spot on the SEM images for a better explanation
8. Figure 12- the legend is not clear
9. Any cross-sectional studies on the composite sample is carried out?
Minor editing of English language required.
Author Response
It can be seen that the authors layout the extensive work with PMMA, HA, and embedded with ZnFe2O4/ZnO. The comments for this manuscript are as follows :
- It is suggested the sub-title for 2.2 is changed to Fabrication instead of Synthesis.
We agree with this recommendation.
- How is the homogeneity of the mixture between PMMA, HA and ZnFe2O4/ZnO is obtained?
We added in Materials and Methods part (marked yellow):
«A solution of PMMA in acetone (10 % wt.) was pre-prepared for uniform mixing. The ZnFe2O4/Zn NPs and HA powder were gradually introduced into the solution. The resulting mixture was thoroughly dispersed using an ultrasonic bath with a frequency of 40±2 kHz (WU-09-«Y-FP»-03, Russia). After removing the solvent, the resulting mixture was pulverized and pellets were obtained».
- What is the basis of selecting the mass percentage of ZnFe2O4/ZnO (5% and 7%)?
We added in Materials and Methods part (marked yellow):
«Based on preliminary studies, it was found that the addition of nanoparticles in the amount of less than 5% did not provide enough antimicrobial activity of the composite»
4.Based on the Figure 8, ion released for Fe3+ is higher than Zn2+, therefore which elemental is mostly induced the antimicrobial activities?
Really, iron ion (Fe3+) release is higher than Zn2+. It was 0.029±0.002 mg/L and 0.034±0.005 mg/L for Zn2+ and Fe3+ cations, respectively.
We added comments in 3.2. Characterization of PMMA based HA/ ZnFe2O4/ZnO composite part (marked yellow):
«According to the values of inhibitory concentrations of iron and zinc ions against, for example, S.aureus, the greatest contribution to antimicrobial activity will be made by zinc ions. The authors [37] was shown that Zn2+ ions effectively kill S.aureus at concentration 10-4 M (0,007 mg/L). At the same time, the Fe3+ ions inhibited the growth of S.aureus ATCC 25923 at concentration 1 mM (56 mg/L) by 80% [38].»
5.Figure 9- label (a) and (b) is not included in the figure
There is a typographical error in the caption of figure 9 (Fig. 10 new). It was corrected, label (a) and (b) were added.
- Figure 10- the legend is not clear
Was changed, marked yellow,
- Figure 11- please locate any significant spot on the SEM images for a better explanation
We added comments in Antimicrobial activity part:
«After 6 hour incubation with the composite sample the bacteria surface became wrinkled and rough (Fig. 12a). The surface of the bacteria becomes rough and some of the bacteria are completely destroyed (Figure 12a, indicate by arrows). At the same time the control S. aureus incubated with PMMA sample have a smooth surface without defects (Figure 12b). Thus, the SEM images demonstrated bacteria morphological changes after composite sample treatment, including cell membrane broken, loss of normal appearance, wrinkled and rough surface. Similar phenomena are also observed in other S.aureus experiments caused by silver nanoparticles [43, 44] and graphene oxide/cobalt ferrite nanoparticles [45]. The results clearly indicated the effect of NPs on the integrity of the bacterial cells».
- Figure 12- the legend is not clear
Was changed.
- Any cross-sectional studies on the composite sample is carried out?
We did the cross-sectional studies using only SEM (see Fig. 7, Fig. 8 new).

Reviewer 2 Report
I thoroughly reviewed the manuscript “New PMMA based hydroxyapatite/ZnFe2O4/ZnO composite with antibacterial performance and low toxicity”. It is very interesting and author presented very good way with a numerous data. My suggestion minor revision
Comments
Abstract line 10 Polymethylmethacrylate (PMAA) include it
Line 15: - prodused to produced
Cytotoxicity tests also corroborated the biocompatibility to the human body. Add the obtained results as author report well data if add the values is much better
Introduction
Mostly related to the work no changes
Material and Methods
2.3. Characterization
Reduce it as don’t give some basic information
Results and Discussion
TEM images – is author done different magnifications, if not, include magnifications in nm
Line 335 remove unwanted text
Antimicrobial and antifungal studies: is author used different concentration of materials or one. Should be mention.
Line 342: Figure 11 a not 10 a and should figure b also
NA
Author Response
1 Abstract line 10 Polymethylmethacrylate (PMAA) include it
Corrected, marked green.
2 Line 15: - prodused to produced
Corrected, marked green.
3 Cytotoxicity tests also corroborated the biocompatibility to the human body. Add the obtained results as author report well data if add the values is much better.
Added, marked green:
«The composite showed excellent biocompatibility against sensitive fibroblast cell line 3T3. The more than 70% cell viability was observed after 1-3 days incubated with sample»
4 Material and Methods
2.3. Characterization
Reduce it as don’t give some basic information
Was reduced.
5 Results and Discussion
TEM images – is author done different magnifications, if not, include magnifications in nm
Magnifications were included.
Was added.
Line 335 remove unwanted text
Removed
6 Antimicrobial and antifungal studies: is author used different concentration of materials or one. Should be mention.
Based on preliminary studies, it was found that the addition of nanoparticles in the amount of less than 5% did not provide antimicrobial activity of the composite. Added in 3.3 antimicrobial activity part (marked green):
Antibacterial activity of PMMA/NPs composites was previously investigated by us in the article [39]. It was found that a mass fraction of nanoparticles in the composite of less than 5% weakly inhibited bacterial growth. This effect is due to the small amount of antibacterial nanoparticles on the surface of the composite.
7 Line 342: Figure 11 a not 10 a and should figure b also
Corrected

Reviewer 3 Report
This is a well-written manuscript that investigates the development of antibacterial bone void filler and other medical implants using PMMA-based nanocomposites. This study is publishable with minor adjustments.
The abstract and introduction of the paper are straightforward and well-written. A general review of the literature describing PMMA, its application in the bone void filler, and the potential drawbacks are provided. The need for developing such antibacterial biomedical fillers is well supported by the literature. The authors were able to synthesize magnetic hybrid ZnO/ZnFe2O4 nanocomposites and further incorporate the nanocomposites into the matrix of PMMA and HA. The materials were shown to have antibacterial performance against various bacteria studied,
This manuscript provides compelling evidence that supports the claims the authors have made, and it is publishable after addressing the following minor issues:
1. Section 3.1, EDS elemental scan shows the existence of mentioned elements. However, to comment on the uniform distribution throughout the particles, EDC mapping overlay with HAADF/STEM/TEM will be needed.
2. For the TEM analysis, is manual drawing around the particles done to measure the size of particles? How many particles were considered within the analysis?
3. The method described in lines 213-216 is not accurate and can not prove the average size of the nanoparticles being around 90nm. Surface area is largely affected by the diameter of the particles, and the TEM suggests very large size distributions.
4. Lines 276-281. The reduction in saturation magnetization can also come from multiple magnetic domains in one composites and aggregates contain multiple particles. Interactions between domains and close particles can cause reduction in saturation magnetization.
Author Response
- Section 3.1, EDS elemental scan shows the existence of mentioned elements. However, to comment on the uniform distribution throughout the particles, EDC mapping overlay with HAADF/STEM/TEM will be needed.
We added EDS mapping on Figure 1 (Fig. 2 new).
- For the TEM analysis, is manual drawing around the particles done to measure the size of particles? How many particles were considered within the analysis?
The particle size was determined by TEM image analysis. We measured the diameter of about 1,700 particles. We added it in Materials and Methods part (2.3. Characterization), marked blue. Usually the number of particles measured is such that the normal logarithmic curve becomes constant
- The method described in lines 213-216 is not accurate and can not prove the average size of the nanoparticles being around 90nm. Surface area is largely affected by the diameter of the particles, and the TEM suggests very large size distributions.
Electrical explosion of conductors is characterized by the fact that the particles do not have a broad particle size distribution. Measurement of particle size by SEM imaging includes procedures for obtaining an ethanol suspension of particles and its deagglomeration by ultrasound. The suspension is then applied to a microscopy grid. The measurement of specific surface area is carried out for dry powder. The closeness of the particle size values obtained by the two methods only confirms the low agglomeration of the particles. This is important for further applications.
- Lines 276-281. The reduction in saturation magnetization can also come from multiple magnetic domains in one composites and aggregates contain multiple particles. Interactions between domains and close particles can cause reduction in saturation magnetization.
We agree with this comment. We added in Results and Discussion Part, marked blue.

Reviewer 4 Report
September, 15th, 2023
Dear authors
Thank you for an interesting report.
In this study, you examined the microbiological properties and biosafety of a composite material created using a new concept. Improving the antibacterial activity of PMMA based hydroxyapatite by incorporating two zinc compounds is expected to greatly contribute to its clinical use. For this reason, I believe that this study will be of great benefit to clinical dentistry and will be an interesting report for the readers of the journal.
I agree with many parts of your claims. However, I think that several revisions are required as follows:
2. Materials and Methods
1. To make it easier for the reader to read this article, I think you should insert a flowchart of the synthesis of ZnFe2O4/ZnO /heterophase nanoparticles and PMMA based HA/ZnFe2O4/ZnO composite.
2. There is no mention of the range of molecular weight or average value of PMMA used for synthesis. I think you had better write the average molecular weight for polymeric materials.
3. Results and discussion
1. In the two pie charts in Figure 1b, the text explaining the composition of each component element is too small and difficult to recognize when viewed on paper. You should bring these descriptions up to a point where they are a little more visible.
Abstract and 4. Conclusions
1. The mouse embryonic fibroblast cell line NIH/3T3, rather than human-derived cells, was used in the biocompatibility experiments in this study. Therefore, the expression ``Cytotoxicity tests also corroborated the biocompatibility to the human body.'' in the Abstract and Conclusions seems overstated. I think you should improve the wording of these parts.

Author Response
- Materials and Methods
- To make it easier for the reader to read this article, I think you should insert a flowchart of the synthesis of ZnFe2O4/ZnO/heterophase nanoparticles and PMMA based HA/ZnFe2O4/ZnO composite.
We added flowchart of the synthesis of ZnFe2O4/ZnO heterophase nanoparticles and PMMA based HA/ZnFe2O4/ZnO composite (Figure 1, «Materials and Methods» part).
- There is no mention of the range of molecular weight or average value of PMMA used for synthesis. I think you had better write the average molecular weight for polymeric materials.
We added this information in «Materials and Methods» part, marked pink.
- Results and discussion
- In the two pie charts in Figure 1b, the text explaining the composition of each component element is too small and difficult to recognize when viewed on paper. You should bring these descriptions up to a point where they are a little more visible.
Figure 1b (new Fig. 2b) was corrected.
Abstract and 4. Conclusions
- The mouse embryonic fibroblast cell line NIH/3T3, rather than human-derived cells, was used in the biocompatibility experiments in this study. Therefore, the expression ``Cytotoxicity tests also corroborated the biocompatibility to the human body.'' in the Abstract and Conclusions seems overstated. I think you should improve the wording of these parts.
We changed Abstract (marked green):
The composite showed excellent biocompatibility against sensitive fibroblast cell line 3T3. The more than 70% cell viability was observed after 1-3 days incubated with sample
We changed Conclusions (marked pink):
In vitro cytotoxicity assay using 3T3 fibroblast cell line showed that all composites were biocompatible, showing >90% cell viability.
